# Identification of differential hypothalamic DNA methylation and gene expression associated with sexual partner preferences in rams

Surajit Bhattacharya[1], Rebecka Amodei[2], Eric Vilain[1], Charles E. Roselli[2]*

**1** Center for Genetic Medicine Research, Children's Research Institute, Children's National Hospital, Washington, DC, United States of America, **2** Department of Chemical Physiology & Biochemistry, Oregon Health & Science University Portland, Portland, Oregon, United States of America

* rosellic@ohsu.edu

**Data Availability Statement:** All FastQC and MultiQC reports are in https://github.com/VilainLab/SheepMethylation/tree/master/FastQC and https://github.com/VilainLab/

## Abstract

The sheep is a valuable model to test whether hormone mechanisms that sexually differentiate the brain underlie the expression of sexual partner preferences because as many as 8% of rams prefer same-sex partners. Epigenetic factors such as DNA methylation act as mediators in the interaction between steroid hormones and the genome. Variations in the epigenome could be important in determining morphological or behavior differences among individuals of the same species. In this study, we explored DNA methylation differences in the hypothalamus of male oriented rams (MORs) and female oriented rams (FORs). We employed reduced representation bisulfite sequencing (RRBS) to generate a genome-wide map of DNA methylation and RNA-Seq to profile the transcriptome. We found substantial DNA methylation and gene expression differences between FORs and MORs. Although none of the differentially methylated genes yielded significant functional terms directly associated with sex development, three differentially expressed genes were identified that have been associated previously with sexual behaviors. We hypothesize that these differences are involved in the phenotypic variation in ram sexual partner preferences, whereas future studies will have to find the specific mechanisms. Our results add an intriguing new dimension to sheep behavior that should be useful for further understanding epigenetic and transcriptomic involvement.

## Introduction

The mechanisms underlying the development of sexual orientation remain unknown. A large amount of empirical data suggest that genes and prenatal hormones are important determinants [1]. Given that sexual orientation represents one of the largest sex differences in humans, the leading neurohormone theory posits that like other sexually dimorphic behaviors, sexual orientation reflects the sexual differentiation of the brain under the influence of androgens. Simply stated, exposure to high levels of androgens during a critical period of gestation (i.e., most males and a few females) programs attraction to females in adulthood. While exposure to

SheepMethylation/tree/master/MultiQC
respectively. All other relevant data are within the
manuscript and its Supporting Information files.

**Funding:** This work was supported by a National
Institutes Health grant R01OD011047 and an
Oregon Health and Science University School of
Medicine Innovation Award both awarded to C.E.R.
The funders had no role in study design, data
collection and analysis, decision to publish, or
preparation of the manuscript. There was no
additional external funding received for this study.

**Competing interests:** The authors have declared
that no competing interests exist.

low levels of androgens (i.e., most females and a few males) programs sexual attraction to males. There is also compelling evidence implicating the involvement of epigenetic mechanisms in mediating the long-term effects of hormones on the sexual differentiation of the brain in animal models [2–4]. Evidence in rodents suggests that perinatal androgen exposure reduces DNA methylation in male brains compared to female brains, releasing masculinizing genes from epigenetic repression and ultimately masculinizing sexual behavior [5] and brain anatomy [6]. It is not known currently whether epigenetic factors influence human sexual orientation although circumstantial evidence suggests that it could [4].

Domestic rams have emerged as an important animal model for human sexual orientation. Approximately 8% of rams in natural populations of common western breeds can be reliably identified to show exclusive and enduring sexual partner preference for either the opposite sex (female-oriented) or same sex (male-oriented) [7]. Like men, rams have a sexually dimorphic nucleus (SDN) in the preoptic area/anterior hypothalamus [8,9]. The volume of the ovine SDN correlates with sexual partner preference and is larger in female-oriented rams than in male-oriented rams and ewes. The precise function of the ovine SDN remains unclear but its volume has been shown to be a biomarker of prenatal androgen exposure [10]. Thus, the volume difference between sexes and between female- and male-oriented rams most likely results from a developmental difference in androgen exposure and may be reflected in differences in DNA methylation states in the brain [11]. The medial basal hypothalamus is another brain area that plays crucial roles in neuroendocrine control systems and sexual behaviors [12]. The ventromedial nucleus is a major anatomical component of the medial basal hypothalamus that is larger in males than in females, regulated by perinatal hormone exposure, and involved in facilitating male sexual behavior [13–16]. The present study evaluated the genome-wide epigenetic and transcriptomic levels of the medial basal hypothalamus in female- and male-oriented rams. We hypothesize that the DNA methylome and transcriptome of the hypothalamus differs between these rams as evidence of a legacy of differential androgen exposure during early fetal development.

## Materials and methods

### Animals and behavioral classifications

Archival hypothalamic tissues were used in this study. The tissue was obtained from 4-5-year-old adult rams that were given behavioral tests at the USDA Sheep Experiment Station in Dubois, ID and classified as male-oriented rams (MORs) (n = 5) or female-oriented rams (FORs) (n = 4). The sheep were of mixed western breeds, including Rambollet, Targhee and Polypay. Rams were given sexual partner preference tests administered as described previously [17]. Those that exclusively mounted other rams were classified as male oriented rams (MORs), whereas rams that exclusively mounted females were classified as female oriented rams (FORs); (Table 1). All experimental animal protocols met the stipulations and guidelines of the NIH policy on the Care and Use of Laboratory Animals and were approved by the Institutional Animal Care and Use Committee of the Oregon Health and Science University.

### Sample collection and preparation

The sheep were euthanized with an overdose (15 mg/kg) of sodium pentobarbital (Euthozol; Delmarval Laboratories Inc, Midlothian, VA). The medial basal hypothalamus was dissected as a block of tissue that extended from the caudal aspect of the optic chiasm to the rostral aspect of the mammillary bodies, bilaterally to the optic nerves and dorsally to the top of the third ventricle. The dissection was split through the ventricle into left and right halves that were frozen immediately on dry ice and stored in a -80˚C freezer. Genomic DNA was extracted from one half of the hypothalamus using the DNeasy Blood & Tissue Kit (Qiagen,

**Table 1. Number of mounts on female and male stimulus animals in the last two of four sexual partner preference tests.**

| Ram Number | Classification | F/M Mounts* SPP Test #3 | F/M Mounts SPP Test #4 |
|---|---|---|---|
| R2579 | MOR/SSP | 0/(36)† | 0/38 |
| R2810 | MOR/SSP | 0/(40) | 0/3 |
| A8337 | MOR/SSP | 0/97 | 0/6 |
| T8384 | MOR/SSP | 0/(56) | 0/(20) |
| R2423 | MOR/SSP | 0/(15) | 0/0 |
| A8736 | FOR/OSP | (7)/0 | (9)/0 |
| R3139 | FOR/OSP | (9)/0 | (8)/0 |
| R3362 | FOR/OSP | (7)/0 | (8)/0 |
| A9707 | FOR/OSP | (6)/0 | (10)/0 |

Germantown, MD, USA) and concentrated using the Genomic DNA Clean & Concentrator Kit (Zymo Research, Irvine, CA, USA) as directed by the manufacturer. The concentration and quality of genomic DNA was verified with absorbance spectroscopy and Qubit fluorimetry (ThermoFisher Scientific, Waltham, MA, USA). RNA was extracted from the remaining half of the hypothalamus using the RNeasy Mini kit (Qiagen). RNA was quantified with the Qubit RNA Broad range kit (Thermofisher) and integrity was verified on a 4200 Tape station (Agilent, Santa Clara, CA, USA). All RNA samples that were used in these studies had RIN values greater than 8.0.

## Reduced representation bisulfite sequencing

To analyze DNA methylation, we used reduced representation bisulfite sequencing (RRBS) [17], a genome-wide approach that examines about 2 million CpGs (7–10% of all CpGs in genome) that are highly enriched key regulatory regions including promoters, CpG islands and CpG island shores.

To generate RRBS libraries, ~150ng of sheep genomic DNA was digested overnight with the restriction enzyme MspI (New England Biolabs, Ipswich, MA, USA). The DNA was then purified with AMPure XP magnetic beads (Beckman Coulter, Pasadena, CA, USA) before use with the NEXTflex Bisulfite-Seq Kit (BioScientifica, Bristol, UK). The DNA was then end repaired, A-tailed and ligated with the NEBNext Methylated Adaptor (New England Biolabs). The ligated DNA was size-selected using AMPure XP magnetic beads to produce a final library size of 350 bp. Bisulfite conversion was performed with the EZ DNA Methylation-Gold Kit (Zymo Research) before carrying out PCR amplification with NEBNext Multiplex Oligos (New England Biolabs) to barcode each library. A final AMPure XP bead purification was performed, and the resulting libraries were quantified with the Qubit High Sensitivity double stranded (dsDNA) Assay (Life Technologies, Carlsbad, CA, USA) and the Bioanalyzer High Sensitivity Analysis (Agilent). Libraries were multiplexed and sequenced on the Illumina Next-Seq or HiSeq2500 to obtain ~30 million single end, 75 bp reads. The sequence data was deposited under the gene expression omnibus (GEO) accession number *GSE158287*. Library names and associated phenotypes are in Table 2.

## Bioinformatic analysis

Quality reports for all the nine sample sequences (five MORs and four FORs), were generated using FastQC [18] (generates per sample quality report) and MultiQC [19] (generates a multi sample quality report, by aggregating the individual FastQC reports). All samples passed the

**Table 2. Sample names and associated Phenotypes.**

| Sample Name | Phenotype |
|---|---|
| LIB181217CR_ECL28_1_S1 | MOR1 |
| LIB181217CR_ECL28_2_S2 | MOR2 |
| LIB181217CR_ECL28_3_S3 | MOR3 |
| LIB181217CR_ECL28_4_S4 | MOR4 |
| LIB181217CR_ECL28_5_S5 | MOR5 |
| LIB181217CR_ECL28_6_S6 | FOR1 |
| LIB181217CR_ECL28_7_S7 | FOR2 |
| LIB181217CR_ECL28_8_S8 | FOR3 |
| LIB181217CR_ECL28_9_S9 | FOR4 |

per base sequence quality metrics, i.e. none of the bases have their lower quartile less than 10 Phred score [20,21] or median less than 25 Phred score (FastQC and MultiQC are in https://github.com/VilainLab/SheepMethylation/tree/master/FastQC and https://github.com/VilainLab/SheepMethylation/tree/master/MultiQC respectively). From the MultiQC reports, it can be observed that for most of the samples the "per base sequence content" graph starts with a C or T followed by two Gs, excepting one sample LIB181217CR_ECL28_7_S7_R1_001 (FOR2) where the percentage of the T's more than Cs or Gs for the first three bases (S1 Fig). This discrepancy in the FOR2 sequence can be due to improper MSPI digestion during library preparation, which in turn does not enrich reads that start with CGG or TGG.

Next, trimming was performed using Trim Galore [22] to get high quality reads for better methylation calls. It trims all reads having a Phred score less than 20 (i.e., 99% base call accuracy), read length less than 20bp after quality trimming and adapter contamination and/or when reads start with CAA or CGA (S1 File). For all but one sample, sequences removed for quality score criterion were less than 15% of the total number of sequences for that sample and for length criterion; it was less than 5%. For LIB181217CR_ECL28_7_S7_R1_001 (FOR2) sample, the sequences removed for quality score criterion were 16.4% and for lengths less than 20 bp were 6.9%. Similarly, for RRBS trimming excluding FOR2, all the samples had RRBS sequences trimmed due to adapter contamination was < 30% and RRBS sequences trimmed due to reads starting with CAA and CGA at 0.1%. For FOR2 sample, the reads trimmed due to adapter contamination were 37%, whereas for the other criteria trimmed reads were 0.2% of all the sequences in the sample (S1 File).

Bismark [23] with the Bowtie 2 [24] alignment option was used to align the trimmed sequence to the reference genome (Oar_rambouillet_v1.0) and extract the methylation pattern, in the form of cytosine reports for 3 different contexts CpG, CHH and CHG (where H can be A, T or C). Default parameters were used for Bismark and the mapping efficiency was between ~64–69%. The percentage of methylated cytosine in CpG sites in the sample, calculated by dividing number of methylated cytosines and total number of CpG sites, ranged from ~45–64% with FOR2 having the maximum methylated cytosines in the CpG context. The range of methylation on C was 0.6–0.8% for all CHH and CHG context (S2 File).

Further quality checks, normalizations and differential methylation analyses were performed using the R Bioconductor package, methylKit [25]. Normalization was performed using the normalizeCoverage function of methylKit, which normalizes the coverage between samples by using a scaling factor derived by the difference of median coverages between samples. Differential methylation can be broadly classified into two parts, differentially methylated cytosines (DMC) and differentially methylated regions (DMR). While DMC looks at differences in methylated cytosines between two conditions (MOR and FOR in this case), the DMR

looks at methylation differences in two regions (non-overlapping 1000 bases in this case) between the conditions. The TileMethylCounts function from the methylKit package was used to estimate the number of methylated Cs in 1000 bases of non-overlapping windows across the whole genome. To identify the number of CpG contexts in a sample, we used a coverage threshold between 10X (i.e., at least 10 reads cover that particular CpG context) and 99[th] percentile of the highest CpG coverage per sample. In addition, at least 3 out of 5 samples were required to pass the coverage criterion. All the samples were then merged by using the unite function in methylKit. A further filtration was applied in this step to remove samples that had at least three replicates having coverage for a CG position. We also merged both the strands to increase the coverage of CpG, using the destrand = TRUE option.

Differential methylation was calculated using calculateDiffMeth and getMethylDi*ff* functions from the methylKit package. The calculateDiffMeth function calculates the differential methylation using a logistic regression model based on a Chi-square test followed by an over-dispersion correction using the McCullagh and Nelder method [26], and then adjusts the p value using a Sliding Linear Model (SLIM) [26] multiple test correction method. The get-MethylDiff function was used to extract the significant hypo and hyper DMR and DMC from the result of calculateDiffMeth function. A false discovery rate (FDR) q value threshold of $< 0.1$ and methylation difference of $\pm 10\%$ was used to identify significant DMRs and DMCs. Annotation of the DMR and DMC was performed using genomation [27]. Codes used for gene alignment, methylation extraction and differential methylation analysis are in https://github.com/VilainLab/SheepMethylation/tree/master/Codes. The workflow of the bioinformatics pipeline for the transcriptomic analysis is illustrated in S2A Fig.

## RNA library preparation and sequencing

Sequencing libraries were prepared using fragmentation, end repair, ligation and PCR using the Ilumina Stranded mRNA Ligation Prep (Ilumina, San Diego, CA, USA). Briefly, 1 ug of total RNA was purified, fragmented and primed with random hexamers to generate first strand complementary DNA (cDNA) and the first stand cDNA was converted into second strand cDNA. The 3' ends of the second strand cDNA were subjected to blunt-end repair. In the next step, pre-index anchors (RNA index anchors) were ligated to the ends of the double-stranded cDNA fragments to prepare them for dual indexing. A subsequent PCR amplification step followed to add the index adapter sequences (IDT for Illumina RNA UD Indexes Set A, Ligation UDP0001-UDP0005). This step selectively amplified the anchor-ligated DNA fragments and adds indexes and primer sequences for cluster generation. For indexing PCR, initial denaturation was carried out at 98˚C for 30 sec, followed by 10 cycles of the following thermal-cycle profile: denaturation at 98˚C for 10 seconds, annealing at 60˚C for 30 seconds, and extension at 72˚C for 30 seconds. A final extension at 72˚C for 5 min was followed by a 4˚C hold. The resulting product was a dual-indexed library of DNA fragments with adapters at each end. The libraries were purified using Agencourt AmPureXP beads (Beckman Coulter) and eluted in 15 μl of resuspension buffer. Libraries were quantified using the Qubit broad range assay kit (Thermofisher) and sized using the DNA 1000 kit (Agilent Technologies). The final 300 bp libraries were pooled in equimolar amounts and normalized. The pooled library (1.2 pM) was sequenced on the Nextseq 550 using the NextSeq 500/550 High Output Kit v2.5 (150 cycles, 2 x 75 bp)and data captured in the Base space sequence Hub (Ilumina).

## RNAseq analysis

Preprocessing of the fastq files were performed using the method mentioned above. Quality check was performed using fastQC [18] for single samples and MultiQC [19] for multi sample

summary, followed by quality and adapter trimming by trimmomatic [22]. Next, the fastq was aligned to the Oar_rambouillet_v1.0 from Ensembl, using STAR [28] followed by read quantification using RSEM [29]. Differential expression analysis was performed using deseq2 [30] with the significance threshold being log2 fold change > 0.58 (1.5 fold change) and log2 fold change < -0.58 (-1.5 fold change), and p-value < 0.1. The workflow of the bioinformatics pipeline for the transcriptomic analysis is illustrated in S2B Fig.

### Functional annotation and visualization

Functional annotation was performed using gProfileR [31]. Visualization was done using methylKit [25], ViewBS [32], ggplot2 [33] and GOplot [34].

### Quantitative PCR method

Total RNA (0.5 μg) was converted to cDNA using the SuperScript™ III First-Strand Synthesis System (Invitrogen, Waltham, MA, USA) according to the manufacturer's directions. Real time PCR reactions were run in triplicate using PowerSYBR Green Master Mix (Invitrogen). Primer sets (S3 File) for ovine genes were designed specifically to cross exon junctions using Clone Manager software version 8 (Sci-Ed Software, Westminster, CO, USA). All reactions were run in a Quant Studio 7 Flex Thermal Cycler (Applied Biosystems, Life Technologies, Eugene, OR, USA). The primer efficiencies were ≥ 95% for all primer pairs, and all melting curves showed a single peak. Quantification of gene expression was performed by the delta delta Ct method, using cDNA from MBH dissections obtained from four adult Polypay rams as calibrators and normalized against the reference gene glyceraldehyde-3-phosphate dehydrogenase (GAPDH). Data are reported as the fold difference relative to the mean for MORs.

## Results

### Distinct differential methylation patterns are observed between MORs and FORs in all the three methylation contexts

A global methylation analysis of the three contexts (CpG, CHG and CHH) for all the samples, reveals higher average methylation levels for CpG context compared to the other 2 contexts (Fig 1A, S5–S11 Figs). For each of the three different methylation contexts investigation of differential methylation patterns between MORs and FORs can be broadly classified into two types: evaluation of differentially methylated cytosines (DMC) and differentially methylated regions (DMR).

The range of CpG context sites, that passes both the coverage, and the sample threshold criterion are between ~985,376 to ~1,201,527 for all samples excepting FOR2, for which the number of CpG context site is 250,828 (Table 3). Hierarchical clustering of the average methylation profile for CpG context revealed that FOR2 was not clustered with the other samples and is an outlier (S4A Fig). The same pattern was observed in the other contexts (CHG and CHH), with FOR2 being an outlier in both the scenarios (S4B and S4C Fig). This makes the FOR2 sample an outlier and it was removed from further downstream DMR and DMC analysis, for all the three contexts.

DNA methylation has various functions, and methylation can occur in different locations. We evaluated 656,897 filtered CpG context sites and identified 1552 DMCs of which 803 were hypomethylated and 749 were hypermethylated in MORs compared to FORs. Of all the differentially methylated cytosines, 44% are located in the intergenic regions, followed by 37% in introns, 11% in exons and 8% in promoters (Fig 2A). A similar distribution pattern of methylated DMCs was observed for hypo- and hypermethylated CpG regions, with the majority of

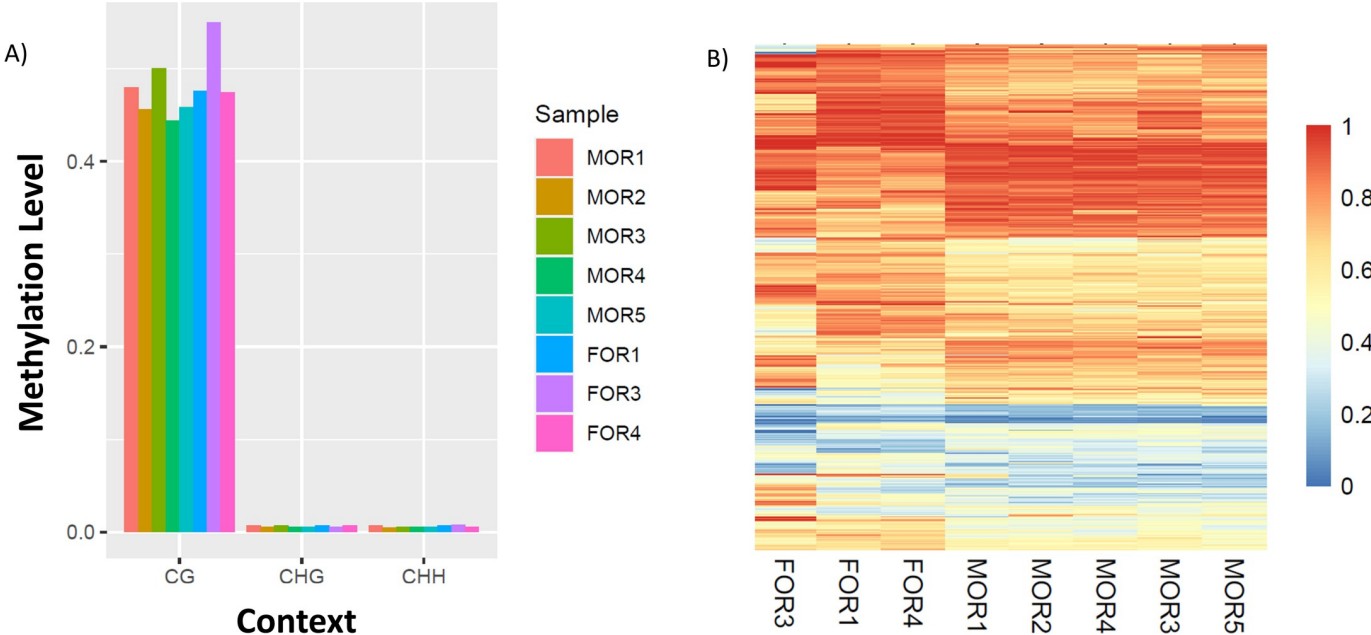

**Fig 1. Distinct global CpG methylation pattern observed between contexts and conditions.** A) Global Methylation levels show higher methylation of CG context: Average Global methylation levels of the eight samples (5 MORs, 3 FORs). CG methylation level is higher than the other contexts for all the samples. B) Heatmap of DMR methylation ratios for reads in the CpG context plotted against animal number shows differential methylation patterns between MORs and FORs. Higher value red, lower value blue.

the DMC's located in intergenic regions and the least in the promoter region (Fig 2B and 2C, S4 File DMC_CpG tab, DMC_hyper_CpG tab and DMC_hypo_CpG tab).

For evaluation of DMRs, we looked at non-overlapping 1000 base pair regions and identified 805 DMRs of which 478 were hypomethylated and 327 were hypermethylated in MORs compared to FORs. Visualization of all DMRs shows distinct differential patterns between the two phenotypes i.e., MORs and FORs (Fig 1B). The distribution of DMRs is similar to that of DMCs across genomic regions with the maximum (46%) falling in the intergenic regions, 33% in introns, 16% in exons and 5% in the promoter regions (Fig 2D). Likewise, the distribution of hypo- and hypermethylated DMRs and DMCs are similar across genomic regions (Fig 2E and 2F, S4 File DMR_CpG tab, DMR_hyper_CpG tab and DMR_hypo_CpG tab).

Although previous studies have not been conclusive about the function of non-CpG (CHG and CHH) methylations in mammals, they have been observed previously in developing

**Table 3. Methylation sites per context.**

| Sample Name | Phenotype | CpG Region | CHG region | CHH region |
|---|---|---|---|---|
| LIB181217CR_ECL28_1_S1 | MOR1 | 1083643 | 1483439 | 2704253 |
| LIB181217CR_ECL28_2_S2 | MOR2 | 1184415 | 1618708 | 3039998 |
| LIB181217CR_ECL28_3_S3 | MOR3 | 1111216 | 1596032 | 3048524 |
| LIB181217CR_ECL28_4_S4 | MOR4 | 1201527 | 1652283 | 3058649 |
| LIB181217CR_ECL28_5_S5 | MOR5 | 1162673 | 1612379 | 3019239 |
| LIB181217CR_ECL28_6_S6 | FOR1 | 1197471 | 1697854 | 3275614 |
| LIB181217CR_ECL28_7_S7 | FOR2 | 250828 | 460853 | 644272 |
| LIB181217CR_ECL28_8_S8 | FOR3 | 1216055 | 1703738 | 3241296 |
| LIB181217CR_ECL28_9_S9 | FOR4 | 985376 | 1367634 | 2531456 |

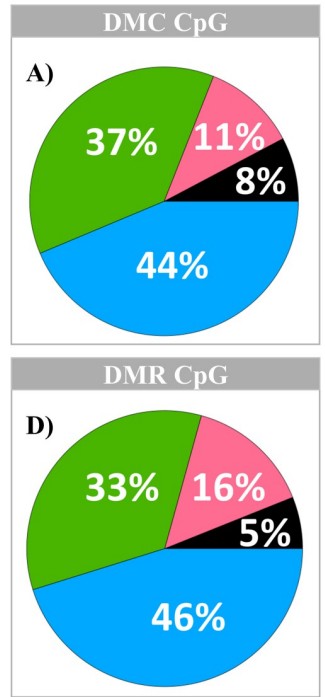
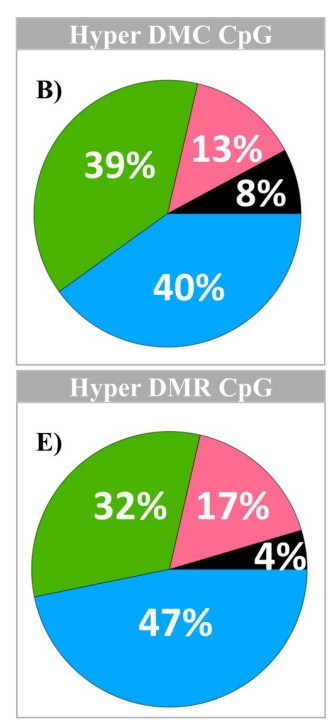
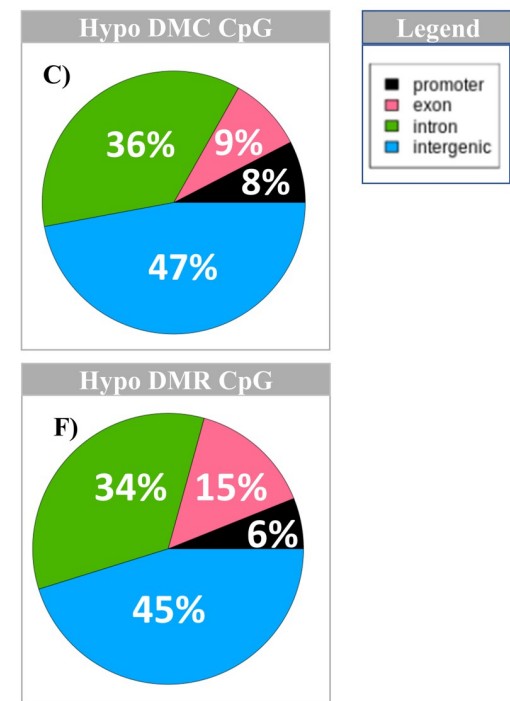

**Fig 2. Distribution of total CpG DMC and DMR across the different genomic regions.** A) Distribution of DMC CpG; B) Distribution of hyper DMC CpG; C) Distribution of hypo DMC CpG; D) Distribution of DMR CpG; E) Distribution of hyper DMR CpG; F) Distribution of hypo DMR CpG; Distribution of hypo DMR CpG. Legend: black = promoter region; pink = exon; green = intron and blue = intergenic regions.

mouse brain [29]. This enabled us to explore the methylation profile in these two contexts. For the CHG context there are 25 DMCs with 10 hypermethylated and 15 hypomethylated (S5 File DMC_CHG tab, DMC_hyper_CHG tab and DMC_hypo_CHG tab), and 16 regions for DMR, with nine hypermethylated and seven hypomethylated. Out of the 25 DMCs, 60% are in the intronic and 40% in intergenic regions (Fig 3A). For the hypermethylated CHGs, 80% are in the intronic and 20% are intergenic regions (Fig 3B), whereas 47% of hypomethylated CHGs fall in intronic and 53% fall in intergenic regions (Fig 3C). For DMR CHGs, 62% fall in intergenic regions, 25% in introns and 12% in exons (Fig 3D). For hypermethylated CHGs, most DMRs (56%) fall in intergenic regions, whereas 33% fall in introns and 11% fall in the exons (Fig 3E). For hypomethylated CHGs, 71% of DMRs fall in intergenic regions, while equal distribution (14%) falls in exons and introns (Fig 3F, S5 File DMR_CHG tab, DMR_hyper_CHG tab and DMR_hypo_CHG tab).

For CHH context, there are 15 DMC regions with 7 hypomethylated and 8 hypermethylated, and 56 DMR regions with 17 hypomethylated and 39 hypermethylated. In the case of CHHs DMCs, the pattern is similar to DMC distribution in CpG context. Out of 15 DMCs, most (60%) fall in intergenic regions, 33% in introns and 7% in promoter regions (Fig 4A). For hypermethylated CHHs, most DMCs fall in intergenic regions (50%), 38% in introns and 12% in exons (Fig 4B). For the hypomethylated CHHs, 71% of the DMCs fall in the intergenic regions and 29% in introns (Fig 4C, S6 File DMC_CHH tab, DMC_hyper_CHH tab and DMC_hypo_CHH tab). The distribution of DMR CHHs follows a similar pattern as DMC, with most falling in intergenic regions (46%), followed by 34% in introns, 14% in exons and 5% in the promoter regions (Fig 4D). Hypomethylated DMRs follow the same pattern as the distribution of all CHH DMRs (most are located in the intergenic regions and the fewest in the

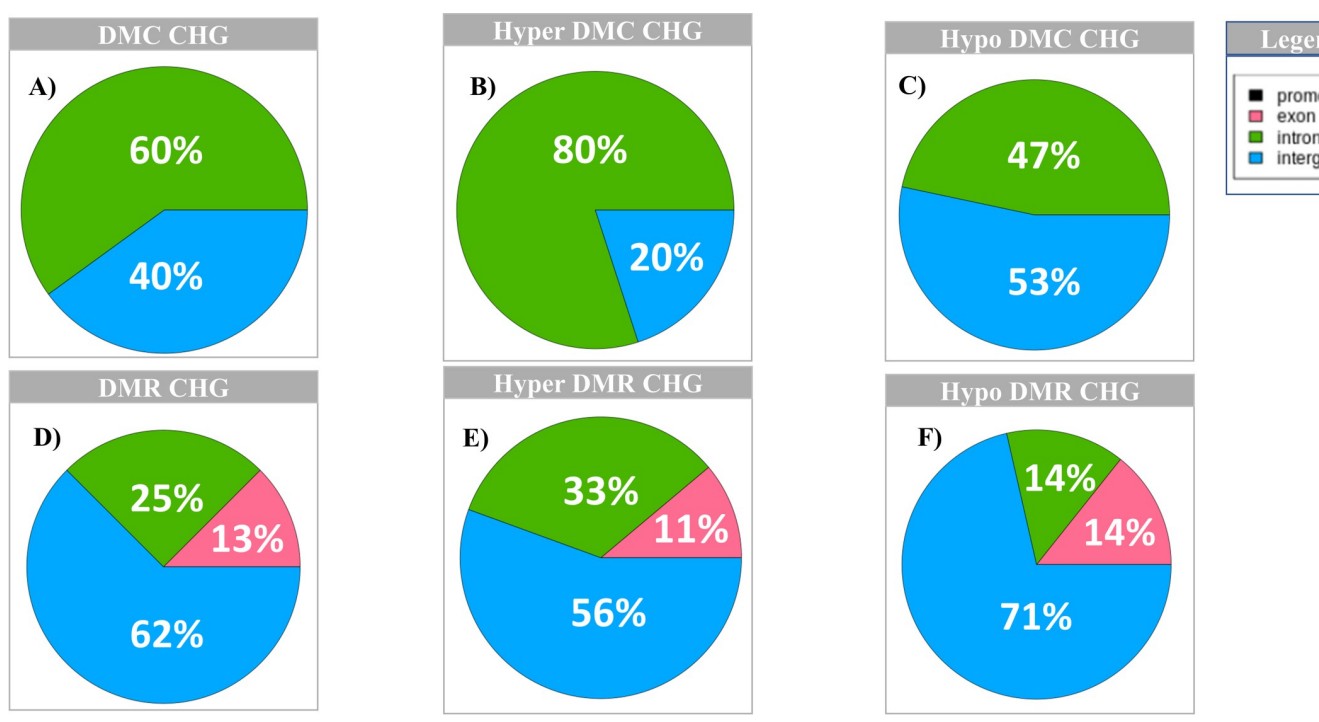

**Fig 3. Distribution of CHG DMC and DMR across the different genomic regions.** A) Distribution of DMC CHG; B) Distribution of hyper DMC CHG; C) Distribution of hypo DMC CHG; D) Distribution of DMR CHG; E) Distribution of hyper DMR CHG; F) Distribution of hypo DMR CHG; Legend: black = promoter region; pink = exon; green = intron and blue = intergenic regions.

promoter region). For hypermethylated DMRs, the same percentage of DMRs fall in the intronic and intergenic regions (38%), followed by exon (16%) and promoter (8%) regions (Fig 4E and 4F, S6 File DMR_CHH tab, DMR_hyper_CHH tab and DMR_hypo_CHH tab).

## Functional annotation of the individual methylation contexts reveals distinct functional clusters

To identify functionally relevant genes overlapping DMC/DMRs, we performed functional annotation using gProfileR. We only chose genes that had DMC/DMRs in their gene body (i.e., exons and introns) or promoters, and left out intergenic DMC/DMRs from further analysis. Functional annotation for DMC CpG (hyper- and hypomethylated cytosines) context produced 28 significantly enriched functional clusters, adjusted p-value < 0.1 (Fig 5A, S4 File DMC_CpG_GO tab). The significantly enriched functional terms are all gene ontology (GO) terms with 14 of them pertaining to biological processes (BP), eight to molecular functions (MF) and six to cellular components (CC). The BP GO clusters contain mainly developmental and biological regulation processes, whereas the MF terms pertains to protein binding and electrophysiological activities, while the CC pathways includes membrane and cytoplasm related terms. The hypermethylated DMCs yielded ten significant GO terms, with seven MF terms comprising electrophysiological and protein binding activities while three CC terms are related to membrane and cation channel complex terms. The hypomethylated DMCs yielded 12 significant functionally relevant terms, with seven BP terms pertaining to biological regulation and response to wound, two MF terms associated with protein binding and three CC terms related to cytoplasm, cell periphery and Schaffer collateral—CA1 synapse (S4 File DMC_ hyper_CpG_GO and DMC_ hypo_ CpG_GO tab).

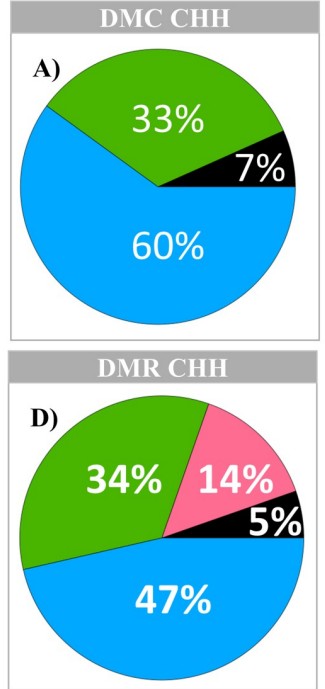
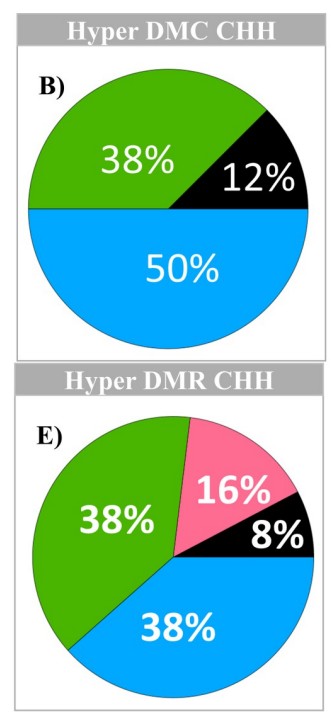
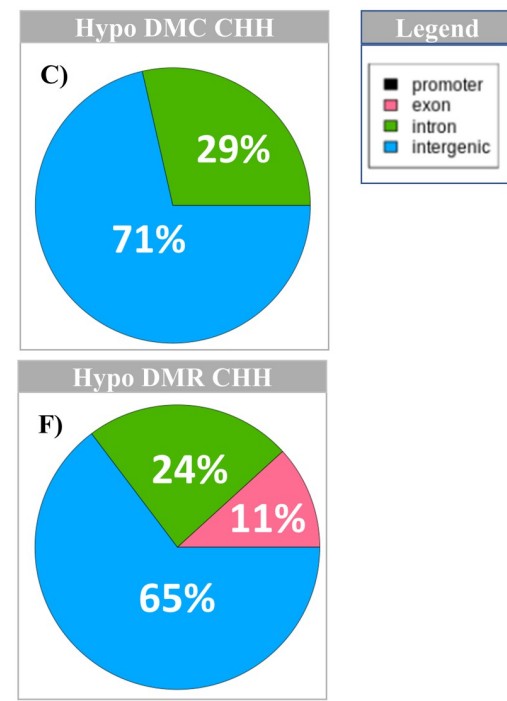
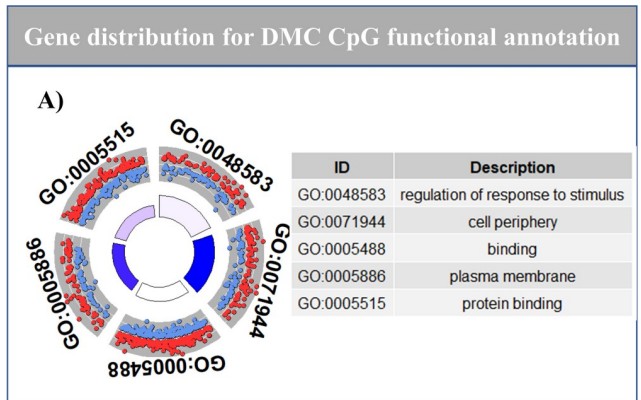

**Fig 4. Distribution of CHH DMC and DMR across the different genomic regions.** A) Distribution of DMC CHH; B) Distribution of hyper DMC CHH; C) Distribution of hypo DMC CHH; D) Distribution of DMR CHH; E) Distribution of hyper DMR CHH; F) Distribution of hypo DMR CHH; Legend: black = promoter region; pink = exon; green = intron and blue = intergenic regions.

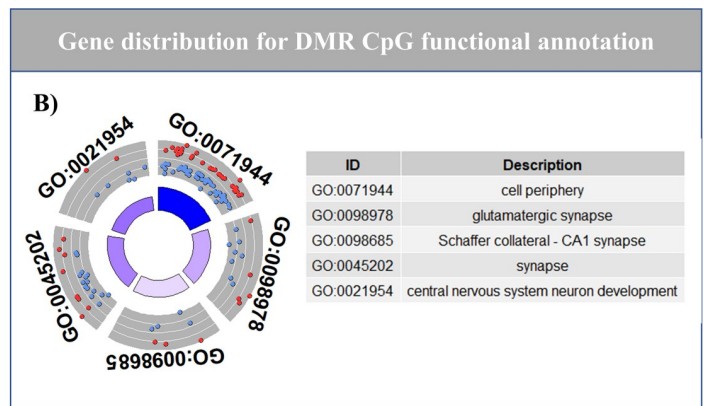
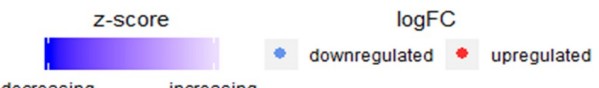

**Fig 5. Functional annotation for DMC and DMR for CpG context.** A) Representation of gene distribution across functional annotation terms for DMC CpG. The distribution of hypermethylated (red) and hypomethylated (blue) terms for each of the functional terms is represented in each quadrant. B) Representation of gene distribution across functional annotation terms for DMR CpG. The distribution of hypermethylated (red) and hypomethylated (blue) terms for each of the functional terms is represented in each quadrant. Enrichment of each term is reported as a z-score, where z-score is the ratio of difference between number of hyper methylated and hypomethylated DMC genes, and square root of total number of genes for that term.

Functional annotation of DMRs for CpG regions revealed nine significant enriched functional terms (Fig 5A, S4 File DMR_CpG_GO tab). Of the nine functional terms, five are CC functions related to synapse and cell periphery, three are BP functions related to central nervous system neuron development, activation of GTPase activity and movement of cell or subcellular component), and one is a MF function associated with calcium ion binding. The hypermethylated DMRs, yielded no significantly enriched terms, whereas the hypomethylated DMRs yielded five terms with three BP functions related to regulation of GTPase activity and chemorepulsion of axons and two MF functions associated with GTPase regulator activity and nucleoside-triphosphatase regulator activity (S4 File DMR_ hyper_CpG_GO and DMR_ hypo_ CpG_GO tab).

The non-CpG methylation yielded fewer functionally relevant terms, compared to the CpG context. The functional annotation for DMCs in CHG context (both hyper- and hypomethylated combined) yielded no significantly enriched terms. Hypermethylated DMCs yielded two significant functionally enriched terms: one human phenotype (unilateral radial aplasia) and one CC function related to mitochondrial pyruvate dehydrogenase complex. Hypomethylated DMCs yielded only one MF function associated with phosphomevalonate kinase activity (S5 File DMC_ CHG_GO tab, DMC_hypo_ CHG_GO tab and DMC_hyper_ CHG_GO tab). The CHG DMRs yielded no significantly enriched functional annotation terms. A similar pattern was observed for functional annotation of DMC/DMR for the CHH context. Only one enriched term was observed for DMCs in the CHH context and it was associated with the MF phosphomevalonate kinase activity. Hypermethylated DMCs yielded three BP functions linked with regulation of clathrin coat assembly and gastric acid secretion. Hypermethylated DMCs yielded one MF term linked to phosphomevalonate kinase activity. Like DMRs in CHG context, none of the DMRs in CHH context yielded any relevant functional terms (S6 File DMC_CHH_GO tab, DMC_hypo_CHH_GO tab, DMC_hyper_CHH_GO tab).

## Overlap of regions across the three methylation contexts show distinct functional features

To understand the effect of the different methylation contexts (CpG, CHH and CHG), we investigated the DMRs that overlap for the multiple contexts. We identified three genes common between all three contexts, one common between CpG and CHG, five common between CpG and CHH and six common between CHH and CHG (Fig 6A; S7 File). The genes common between the three contexts are *ENSOARG00020023439*, *TPGS2* and *SCNN1B*. The DMR was in an intergenic region near *ENSOARG00020023439* (spindlin-2B homologue in sheep; DMR coordinates: chromosome X- 50,554,001–50,555,000; intergenic near the gene) and was hypomethylated (methylation difference MD = -35.6%, corrected P = 0.007) in MOR compared to FOR in the CpG context, whereas it was hypermethylated in the CHG (MD = 16.36%, P = 0.08) and CHH context (MD = 16.6%, P = 0.05). For *TPGS2* (tubulin polyglutamylase complex subunit; chromosome 23: 24,612,001–24,613,000), the DMR was in the intron regions and was hypermethylated in MOR compared to FOR, in all the three contexts (CpG: MD = 30.15%, P = 0.07; CHG: MD = 10.63%, P = 0.06; CHH: MD = 11.14%, P = 0.005). A similar pattern was observed for the gene *SCNN1B* (sodium channel epithelial one subunit beta, chromosome 24: 21,728,001–21,729,000), with the DMR occurring in exon and intron regions, and hypermethylated for all the three contexts (CpG: MD = 14.4%, P = 0.08; CHG: MD = 15.75%, P = 0.03; CHH: MD = 15.86%, P = 0.0001; S7 File). The three genes also have distinct functional features. While spindlin-2B (human homologue of *ENSOARG00020023439*), is involved in regulation of cell cycle progression [35] and H3K4me3-binding activity [36], *TPGS2* codes for a protein component of neuronal polyglutamylase complex [37], whereas *SCNN1B* is responsible for sodium

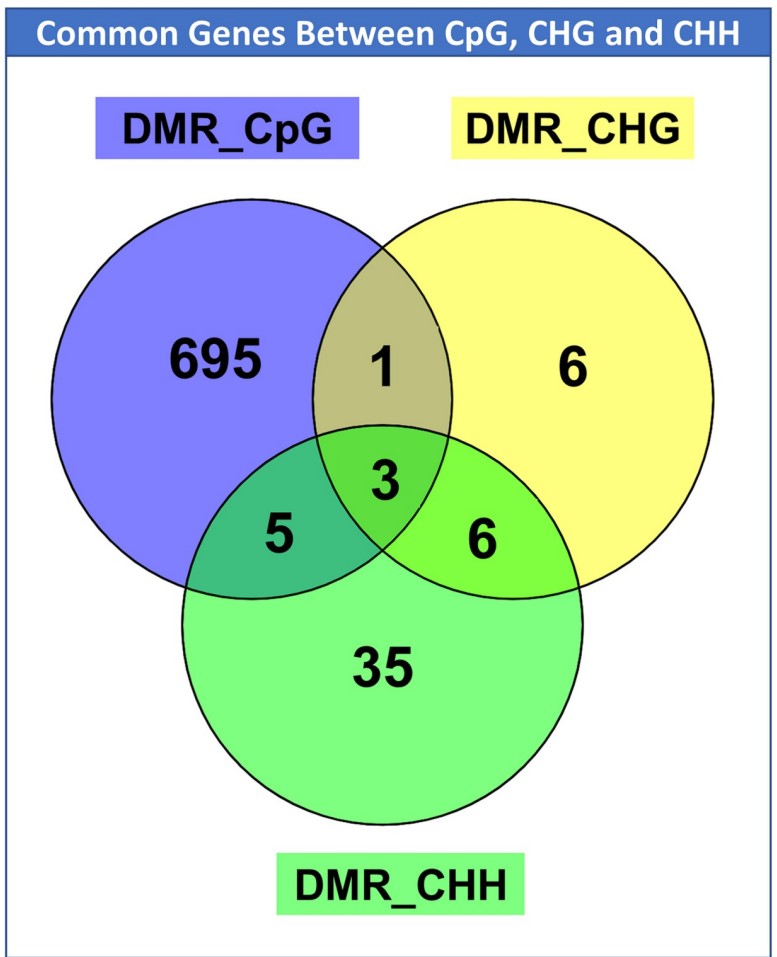

**Fig 6. DMR genes common between the three contexts.** DMR genes common between the three contexts. Venn diagram depicting the genes that are shared among CHH (green), CpG(purple) and CHG (yellow), in the DMC context. There are three genes in common among the 3 contexts, one in common between CpG and CHG, five in common between CpG and CHH, and six in common between CHG and CHH.

channel activity and mutation of the gene leads to autosomal disorders like Liddle syndrome [38].

The DMR for the unannotated gene ENSOARG00020011386 (DMR coordinates: chromosome 18: 66,982,001–66,983,000) common between CHG and CpG contexts was in the intergenic region near the gene and was hypomethylated in MORs for both the contexts (S7 File). For the six genes common between CHH and CHG, the DMR for four genes (ENSOARG00020000663, EPCAM, ADAMTS15, and PLXND1) were in the intergenic region, whereas for the other two genes (MAGI1, TVP23A) the DMR was in the intronic region. All the genes except one (PLXND1), was hypermethylated in MORs compared to FORs, in the two contexts. Functional annotation of the genes revealed three significantly enriched CC functional terms related to cellular junctions. DMRs for the genes common between CHH and CpG, overlap the gene body with four genes (*U6*, *GSE1*, *MIR153-2* and *AGPAT4*), having DMRs in the intron, whereas for *CARD11*, DMR overlaps both exon and intron. There were only two significant functional annotation terms, one CC (CBM complex) and one HP (decreased specific antibody response to polysaccharide vaccine) associated with these genes (S6 File).

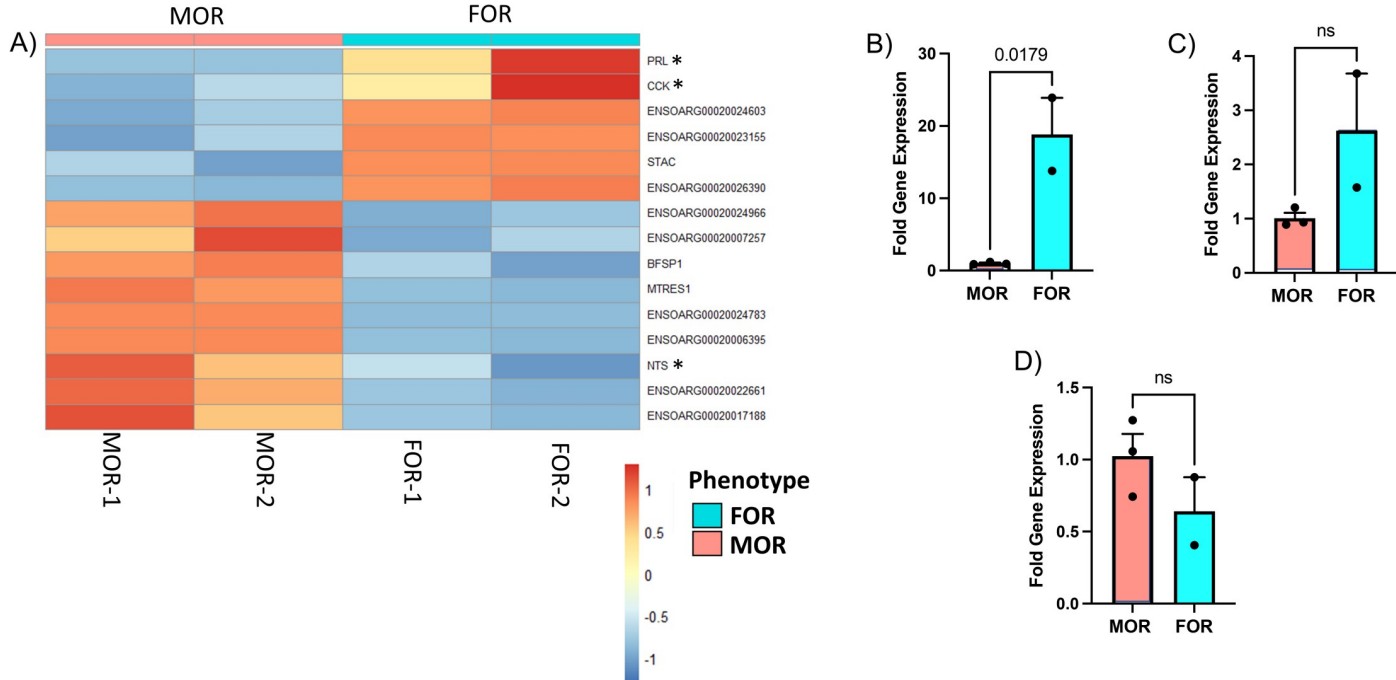

**Fig 7. Differential gene expression associated with sexual partner preference phenotype.** A) Heatmap of differentially expressed genes plotted against animal number and grouped by phenotype, i.e., FOR or MOR. RNAseq analysis identified 15 genes that were differentially expressed between FORs and MORs (adjusted p-value < 0.1, log2 Fold change ≥ absolute (0.58). Heatmap colors are represented by Z-score and annotation of ram phenotype has blue for FORs and red for MORs. Go pathway analysis identified enrichment of three differentially expressed genes involved in hormone activity: prolactin (*PRL*); MOR vs. FOR log2 fold difference (log2 FD) = -4.5, P = 1.8E-07), cholecystokinin (*CCK*); log2 FD = -1.2, P = 5.09E-05 and neurotensin (*NTS*); log2 FD = 1.4, P = 8.40E-06. Differential gene expression was confirmed using qPCR for: (B) *PRL* (log2 FD = -4.2, P = 1.8E-02), (C) *CCK* (log2 FD = -1.37, P = 0.13) and (D) *NTS* (log2 FD = 0.67, P = 0.24). Data (mean ± SEM) were analyzed by a Student's t test.

## Differential expression analysis reveals significantly expressed genes associated with sexual partner preference

To investigate the relationship between DNA methylation changes and gene expression, RNA-Seq analysis was performed to identify differences in gene expression between the two phenotypes. A total of 15 differentially expressed genes were detected between phenotypes, with only one gene overlapping with the DMR gene lists and none with the DMC gene lists (Fig 7A). The gene *BFSP1* (log2 FC = 1.15, qvalue = 0.002), was hypomethylated in CpG DMR context, with a methylation difference of -11.6% MOR vs. FOR (q value = 0.02; S8 Table in S8 File).

To explore further the effect of the differentially expressed genes, we performed functional annotation. GO pathway analysis identified enrichment of three genes involved in hormone activity (MF): prolactin (*PRL*); MOR vs. FOR log2 fold difference (log$_2$ FD) = -4.5, Fold change = -0.04, P = 1.8E-07), cholecystokinin (*CCK*); log$_2$ FD = -1.2, Fold change = -0.43, P = 5.09E-05 and neurotensin (*NTS*); log$_2$ FD = 1.4, Fold Change = 2.639016, P = 8.40E-06 (S8 File). To confirm the differences in gene expression between MORs and FORs identified with RNAseq, we performed quantitative real-time PCR. We observed down regulation of *PRL in MORs vs. FORs* (log$_2$ FD = -4.2, Fold Change = - 0.054, P = 0.001) and *CCK* (log$_2$ FD = -1.3, Fold Change = -0.406, P = 0.08) and up regulation of NTS (log$_2$ FD = 0.67, Fold Change = 1.59, P = 0.24) in MORs compared to FORS, which is in accordance with what was seen in the RNA-Seq analysis (Fig 7B and 7C).

## Discussion

In the present study, genome-wide DNA methylation in hypothalami of rams exhibiting exclusive male versus female sexual partner preferences were analyzed for the first time. Out of the three methylation contexts, CpG, CHG and CHH, the most significant differences were observed in the CpG context with 1552 DMC and 805 DMRs being significantly methylated. There were more hypomethylated CpGs in MORs compared to FORs for both the DMC and DMR groups. The distribution in the case for DMCs was ~52% hypomethylated and ~48% hypermethylated, whereas for DMRs the distribution was 60% hypomethylated compared to 40% hypermethylated. Functional annotation of the differentially methylated genes that fall in the DMC or DMR regions revealed that most of the significant functional terms were related to developmental processes, regulatory and electrophysiological activities that may be associated with the many homeostatic functions of the hypothalamus. Functional terms associated with development of sexual characteristics and sex development were also identified but none of them was differentially enriched.

CpG is considered the most relevant context because 80% of methylation events in humans occur at CpG sites [39]. However, we also evaluated the CHG and CHH contexts because they have been previously associated with brain development [40]. Moreover, CHH methylation is highly conserved in the brain across vertebrate species and requires active maintenance in postmitotic neurons [41]. We observed a pattern similar to previous studies [39,41], with fewer significantly methylated DMCs and DMRs in the non-CpG (CHH and CHG) context, compared to the CpG context. The DMCs for the CHG context followed the same pattern as for CpG, with more hypomethylated than hypermethylated genes. In contrast, the DMRs for the CHG context, and both DMC and DMR for the CHH contexts, exhibited more hypermethylated than hypomethylated genes. There were only a few significant functional terms in both the contexts, and most of them were related to molecular functions such as phosphomevalonate kinase activity or biological processes pertaining to regulation of clathrin coat assembly and regulation of gastric acid secretion. There were genes in common among all three different contexts. Most of them were associated with molecular functions and cellular component functionalities and none was associated with sexual behaviors, neuroendocrine functions or development.

Transcriptomic analysis revealed 15 differentially expressed genes between the two phenotypes with only one overlapping with the methylated list. The gene, Beaded Filament Structural Protein 1 (*BFSP1*) was hypomethylated in CpG DMR context and overexpressed in MORs compared to FORs. This gene shows broad expression in a number of tissues including brain, and has been previously associated with cataracts in humans [42–44]. Additionally, functional annotation of the differentially expressed genes reveal one significant term associated with hormone activity (MF) and consisting of three genes PRL, CCK and NTS. Prolactin (PRL) is a hormone produced mainly by the pituitary gland, however, in some species it is synthesized in other tissues including brain [45,46]. PRL is best known for its role in the development of the mammary gland and milk production, but is also involved in the regulation of parental and sexual behaviors in both males and females [47,48]. The neuropeptide cholecystokinin (CCK) has been associated with mate preference in mice. CCK-expressing neurons in the bed nucleus of the stria terminalis of males are activated by the scent of female urine in association with the male's preference for estrus females [49]. Finally, neurotensin (NTS*)* neurons in the medial preoptic area were shown to encode attractive male cues and direct behavior toward opposite-sex conspecifics in both sexes to drive social attraction toward a potential mate [50]. Quantitative PCR validations show, that PRL is the only gene that was significantly downregulated in MORs compared to FORs, which agrees with the RNASeq results. Although, neither CCK nor

NTS showed significant fold differences with quantitative PCR, they show similar trends with RNA-Seq results, i.e., CCK downregulated and NTS upregulated in MORs compared to FORs.

To our knowledge, our study presents the first genome-wide analysis of DNA methylation profiles and gene expression of the adult sheep hypothalamus. We show that the epigenome of the hypothalamus, in the form of DNA methylation pattern, differs substantially between rams with different sexual partner preferences. This tentatively suggests that epigenetic factors may be important mechanisms involved in sexual attraction. Specifically, we highlight expression differences in genes related to sexual behaviors. These data will be informative in providing a basis for better understanding of the epigenetic regulation of sexual behavior in sheep and help ascertain mechanisms that shape sexual partner preferences. However, further studies will be required to determine whether differences in DNA methylation and consequent gene expression are the cause or consequence of altered behavior. In addition, experiments should be conducted at earlier developmental landmarks are needed to capture the effects more efficiently.

Finally, an important limitation of the present study is its relatively small sample, which limits the statistical power of our conclusions. Sample size is often a challenge with large animal models such as the sheep. This study is no exception and would benefit from a replication with more animals. Thus, further transcriptomic and epigenetic studies need to be performed with a larger sample size to better ascertain the developmental effect that the epigenome/transcriptome has on the expression of sexual partner preferences in rams.

## Supporting information

**S1 Fig. Per base sequence content graph shows discrepancy in starting bases of the sequence in FOR7 compared to other samples.** A) Sequence content across all bases for FOR1: Graph showing the representation of the nucleotides across all base position in sample FOR1. Percentage of C (marked in blue) is highest in the first base followed by G (in black) in the next two positions. Similar pattern was observed in all other samples, excepting FOR2. B) Sequence content across all bases for FOR2: Graph showing the representation of the nucleotides across all base position in sample FOR2. Percentage of T (marked in red) is highest in the first three base. Color code Thymine (T) = red, Adenosine = Green, Cytosine = Dark Blue, Guanine = Black.
(PDF)

**S2 Fig. Workflow of methylation analysis pipeline.** Quality Check using fastqc and trimgalore was used to trim reads less than 20 Phred score. Alignment and methylation count was calculated using Bismark, followed by methylKit to estimate the differentially methylated regions (DMRs) and differentially methylated cytosines (DMCs). Methylation fold change greater than 10; and q value < 0.01 was used for determining the most significant Genes. Annotation of the DMR and DMC was done using genomation. Functional annotation was performed using gProfiler functional annotation tool; followed by visualization using ViewBS for heatmaps, methylKit for dendrogram and distribution of genomic regions for DMRs and GOPlot for gene ontology visualization.
(PDF)

**S3 Fig. Quality of the sample reveals, sample 2810 has more reads than the other samples.** A) Raw read counts from fastq: The plot of the sequence counts shows that for the trimmed sample 2810, the number of reads is greater than 175 million reads, whereas for the other samples has 30 to 75 million reads. We can also observe that the number of duplicate reads (black) in this sample is also greater than the other samples. C) Align read counts from STAR: Aligned read counts from STAR show that sample 2810 has more unmapped reads (red), and least

uniquely mapped reads (dark blue) than any of the other samples.
(PDF)

**S4 Fig. Hierarchical clustering of sample methylation patterns across the 3 contexts.** A) CpG hierarchical Clustering: Hierarchical clustering of the methylation pattern of replicates of all the samples, in CpG context) CHG hierarchical Clustering: Hierarchical clustering of the methylation pattern of replicates of all the samples, in CHG context. C) CpG hierarchical Clustering: Hierarchical clustering of the methylation pattern of replicates of all the samples, in CHH context.
(PDF)

**S5 Fig. Methylation pattern of chromosome 1 and chromosome 2 in CG, CHG and CHH context.** Average methylation levels of the different context between the 2 different samples MOR (red) and FOR (blue). Y-axis average methylation levels, x-axis chromosome coordinates in mega base (Mb).
(PDF)

**S6 Fig. Methylation pattern of chromosome 3 and chromosome 4 in CG, CHG and CHH context.** Average methylation levels of the different context between the 2 different samples MOR (red) and FOR (blue). Y-axis average methylation levels, x-axis chromosome coordinates in mega base (Mb).
(PDF)

**S7 Fig. Methylation pattern of chromosome 5 and chromosome 6 in CG, CHG and CHH context.** Average methylation levels of the different context between the 2 different samples MOR (red) and FOR (blue). Y-axis average methylation levels, x-axis chromosome coordinates in mega base (Mb).
(PDF)

**S8 Fig. Methylation pattern of chromosome 7 and chromosome 8 in CG, CHG and CHH context.** Average methylation levels of the different context between the 2 different samples MOR (red) and FOR (blue). Y-axis average methylation levels, x-axis chromosome coordinates in mega base (Mb).
(PDF)

**S9 Fig. Methylation pattern of chromosome 9 and chromosome 10 in CG, CHG and CHH context.** Average methylation levels of the different context between the 2 different samples MOR (red) and FOR (blue). Y-axis average methylation levels, x-axis chromosome coordinates in mega base (Mb).
(PDF)

**S10 Fig. Methylation pattern of chromosome 11 and chromosome 12 in CG, CHG and CHH context.** Average methylation levels of the different context between the 2 different samples MOR (red) and FOR (blue). Y-axis average methylation levels, x-axis chromosome coordinates in mega base (Mb).
(PDF)

**S11 Fig. Methylation pattern of chromosome 13 and chromosome 14 in CG, CHG and CHH context.** Average methylation levels of the different context between the 2 different samples MOR (red) and FOR (blue). Y-axis average methylation levels, x-axis chromosome coordinates in mega base (Mb).
(PDF)

**S12 Fig. Methylation pattern of chromosome 15 and chromosome 16 in CG, CHG and CHH context.** Average methylation levels of the different context between the 2 different samples MOR (red) and FOR (blue). Y-axis average methylation levels, x-axis chromosome coordinates in mega base (Mb).
(PDF)

**S13 Fig. Methylation pattern of chromosome 17 and chromosome 18 in CG, CHG and CHH context.** Average methylation levels of the different context between the 2 different samples MOR (red) and FOR (blue). Y-axis average methylation levels, x-axis chromosome coordinates in mega base (Mb).
(PDF)

**S14 Fig. Methylation pattern of chromosome 19 and chromosome 20 in CG, CHG and CHH context.** Average methylation levels of the different context between the 2 different samples MOR (red) and FOR (blue). Y-axis average methylation levels, x-axis chromosome coordinates in mega base (Mb).
(PDF)

**S15 Fig. Methylation pattern of chromosome 21 and chromosome 22 in CG, CHG and CHH context.** Average methylation levels of the different context between the 2 different samples MOR (red) and FOR (blue). Y-axis average methylation levels, x-axis chromosome coordinates in mega base (Mb).
(PDF)

**S16 Fig. Methylation pattern of chromosome 23 and chromosome 24 in CG, CHG and CHH context.** Average methylation levels of the different context between the 2 different samples MOR (red) and FOR (blue). Y-axis average methylation levels, x-axis chromosome coordinates in mega base (Mb).
(PDF)

**S17 Fig. Methylation pattern of chromosome 25 and chromosome 26 in CG, CHG and CHH context.** Average methylation levels of the different context between the 2 different samples MOR (red) and FOR (blue). Y-axis average methylation levels, x-axis chromosome coordinates in mega base (Mb).
(PDF)

**S18 Fig. Methylation pattern of chromosome XX in CG, CHG and CHH context.** Average methylation levels of the different context between the 2 different samples MOR (red) and FOR (blue). Y-axis average methylation levels, x-axis chromosome coordinates in mega base (Mb).
(PDF)

**S1 File. Results of quality trimming step by TrimGalore.**
(XLSX)

**S2 File. Results from the alignment and methylation sites determination steps.**
(XLSX)

**S3 File. Oligonucleotide primers used for real-time polymerase chain reaction.**
(XLSX)

**S4 File. Differentially methylation and Functional annotation of CpG context.**
(XLSX)

**S5 File. Differentially methylation and Functional annotation of CHG context.**
(XLSX)

**S6 File. Differentially methylation and Functional annotation of CHH context.**
(XLSX)

**S7 File. DMR Genes Overlapping between CpG, CHG and CHH context.**
(XLSX)

**S8 File. Differentially expressed genes between MORs and FORs.**
(XLSX)

## Acknowledgments

The reduced-representation bisulfite sequencing libraries were generated by the Knight Cardiovascular Research Institute Epigenetics Consortium at OHSU. The authors wish to thank Dr. Lucia Carbone for her oversight of the bisulfite sequencing and helpful comments on the manuscript. The authors would like to thank Dr. Susan Knoblach and Karuna Panchapakesan, for their help with the RNA sequencing.

## Author Contributions

**Conceptualization:** Eric Vilain, Charles E. Roselli.

**Data curation:** Surajit Bhattacharya.

**Formal analysis:** Surajit Bhattacharya, Eric Vilain, Charles E. Roselli.

**Funding acquisition:** Charles E. Roselli.

**Investigation:** Rebecka Amodei.

**Methodology:** Rebecka Amodei.

**Project administration:** Charles E. Roselli.

**Resources:** Eric Vilain, Charles E. Roselli.

**Validation:** Surajit Bhattacharya.

**Visualization:** Surajit Bhattacharya.

**Writing – original draft:** Surajit Bhattacharya, Eric Vilain, Charles E. Roselli.

**Writing – review & editing:** Surajit Bhattacharya, Rebecka Amodei, Eric Vilain, Charles E. Roselli.

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
