## [Decision Letter · Decision Letter 0]

3 Mar 2022

PONE-D-22-01111Identification of differential hypothalamic DNA methylation and gene expression associated with sexual partner preferences in ramsPLOS ONE

Dear Dr. Roselli,

Thank you for submitting your manuscript to PLOS ONE. After careful consideration, we feel that it has merit but does not fully meet PLOS ONE’s publication criteria as it currently stands. Therefore, we invite you to submit a revised version of the manuscript that addresses the points raised during the review process.

The reviewer and academic editor found the submission to be of great interest and believe it provides a meaningful contribution to our our understanding of the hypothalamus as it relates to sexual preference. However, both the reviewer and I agree that a limitation of the study is its relatively smaller sample size. As such, the manuscript would benefit from recognizing its sample size as a potential limitation of the study, perhaps with a brief statement in discussion.  Please submit your revised manuscript by Apr 17 2022 11:59PM. If you will need more time than this to complete your revisions, please reply to this message or contact the journal office at plosone@plos.org. Please include the following items when submitting your revised manuscript:A rebuttal letter that responds to each point raised by the academic editor and reviewer(s). You should upload this letter as a separate file labeled 'Response to Reviewers'.A marked-up copy of your manuscript that highlights changes made to the original version. You should upload this as a separate file labeled 'Revised Manuscript with Track Changes'.An unmarked version of your revised paper without tracked changes. You should upload this as a separate file labeled 'Manuscript'.If applicable, we recommend that you deposit your laboratory protocols in protocols.io to enhance the reproducibility of your results. Protocols.io assigns your protocol its own identifier (DOI) so that it can be cited independently in the future. For instructions see: https://journals.plos.org/plosone/s/submission-guidelines#loc-laboratory-protocols. Additionally, PLOS ONE offers an option for publishing peer-reviewed Lab Protocol articles, which describe protocols hosted on protocols.io. Read more information on sharing protocols at https://plos.org/protocols?utm_medium=editorial-email&utm_source=authorletters&utm_campaign=protocols.

We look forward to receiving your revised manuscript.

Kind regards,

Juan M Dominguez, PhD

Academic Editor

PLOS ONE

Journal Requirements:

(This work was supported by a National Institutes Health grant R01OD011047 and an Oregon Health and Science University School of Medicine Innovation Award both awarded to C.E.R. The funders had no role in study design, data collection and analysis, decision to publish, or preparation of the manuscript.)

3.Thank you for stating the following in the Acknowledgments Section of your manuscript: 

(Acknowledgements: The reduced-representation bisulfite sequencing libraries were generated by the Knight Cardiovascular Research Institute Epigenetics Consortium at OHSU. The authors wish to thank Dr. Lucia Carbone for her oversight of the bisulfite sequencing and helpful comments on the manuscript. The authors would like to thank Dr. Susan Knoblach and Karuna Panchapakesan, for their help with the RNA sequencing. This work was supported by National Institutes Health grants R01OD011047 and OHSU SOM Innovation Award to C.E.R.)

(This work was supported by a National Institutes Health grant R01OD011047 and an Oregon Health and Science University School of Medicine Innovation Award both awarded to C.E.R. The funders had no role in study design, data collection and analysis, decision to publish, or preparation of the manuscript.)

Reviewers' comments:

Reviewer's Responses to Questions

**Comments to the Author**

1. Is the manuscript technically sound, and do the data support the conclusions?

Reviewer #1: Yes

2. Has the statistical analysis been performed appropriately and rigorously? 

Reviewer #1: Yes

3. Have the authors made all data underlying the findings in their manuscript fully available?

Reviewer #1: Yes

4. Is the manuscript presented in an intelligible fashion and written in standard English?

Reviewer #1: Yes

5. Review Comments to the Author

Reviewer #1: An interesting and potential important study examining gene expression in association with sexual preference in sheep. While it would be stronger with a higher n for both female and male oriented males and in the future with a design that perhaps has the researchers blinded to the orientation of the ram the results provided generate important discussion and direction for future research. The authors point out that while is a significant difference in the degree of methylation there are not specific genes associated with sex. I would say that this is not surprising and does not reduce the value of the findings. The manuscript is well-written.

6. PLOS authors have the option to publish the peer review history of their article (what does this mean?). If published, this will include your full peer review and any attached files.

Reviewer #1: **Yes: **Bruce S. Cushing

---

## [Author Response · Author response to Decision Letter 0]

18 Mar 2022

1. We agree that the sample size is a limitation of our study. Although, we commented on this in our original discussion, we added an additional statement to more directly alert the reader to this weakness. See L499-500.

2. As requested, we removed the funding statement from our Acknowledgements.

3. Our updated funding statement should read: “This work was supported by a National Institutes Health grant R01OD011047 and an Oregon Health and Science University School of Medicine Innovation Award both awarded to C.E.R. The funders had no role in study design, data collection and analysis, decision to publish, or preparation of the manuscript. There was no additional external funding received for this study.

4. We provided the GEO accession number (GSE158287) in the text (Line 120) and will put a release date on the GEO public access once we get an acceptance.

5. We checked/resaved our figures using PACE

---

## [Editor Report · Decision Letter 1]

14 Apr 2022

Identification of differential hypothalamic DNA methylation and gene expression associated with sexual partner preferences in rams

PONE-D-22-01111R1

Dear Dr. Roselli,

We’re pleased to inform you that your manuscript has been judged scientifically suitable for publication and will be formally accepted for publication once it meets all outstanding technical requirements.

Kind regards,

Juan M Dominguez, PhD

Academic Editor

PLOS ONE

---

## [Editor Report · Acceptance letter]

26 Apr 2022

PONE-D-22-01111R1 

Identification of differential hypothalamic DNA methylation and gene expression associated with sexual partner preferences in rams. 

Dear Dr. Roselli:

I'm pleased to inform you that your manuscript has been deemed suitable for publication in PLOS ONE. Congratulations! Your manuscript is now with our production department. 

Kind regards, 

on behalf of

Dr Juan M Dominguez 

Academic Editor

PLOS ONE